# Validation of a Bacteriophage Hide Application to Reduce STEC in the Lairage Area of Commercial Beef Cattle Operations

**DOI:** 10.3390/foods12234349

**Published:** 2023-12-01

**Authors:** Makenzie G. Flach, Onay B. Dogan, Mark F. Miller, Marcos X. Sanchez-Plata, Mindy M. Brashears

**Affiliations:** International Center for Food Industry Excellence, Department of Animal and Food Sciences, Texas Tech University, Lubbock, TX 79409, USA; maflach@ttu.edu (M.G.F.); onay.dogan@huskers.unl.edu (O.B.D.); markus.miller@ttu.edu (M.F.M.); marcos.x.sanchez@ttu.edu (M.X.S.-P.)

**Keywords:** pre-harvest, bacteriophage, Shiga toxin-producing *E. coli*, *Salmonella*, cattle hides, boot swabs, lairage area

## Abstract

Finalyse, a T4 bacteriophage, is a pre-harvest intervention that utilizes a combination of bacteriophages to reduce incoming *Escherichia coli* O157:H7 prevalence by destroying the bacteria on the hides of harvest-ready cattle entering commercial abattoirs. The objective of this study was to evaluate the efficacy of Finalyse, as a pre-harvest intervention, on the reduction in pathogens, specifically *E. coli* O157:H7, on the cattle hides and lairage environment to overall reduce incoming pathogen loads. Over 5 sampling events, a total of 300 composite hide samples were taken using 25 mL pre-hydrated Buffered Peptone Water (BPW) swabs, collected before and after the hide wash intervention, throughout the beginning, middle, and end of the production day (*n* = 10 swabs/sampling point/timepoint). A total of 171 boot swab samples were also simultaneously taken at the end of the production day by walking from the front to the back of the pen in a pre-determined ‘Z’ pattern to monitor the pen floor environment from 3 different locations in the lairage area. The prevalence of pathogens was analyzed using the BAX^®^ System Real-Time PCR Assay. There were no significant reductions observed for *Salmonella* and/or any Shiga toxin-producing *E. coli* (STEC) on the hides after the bacteriophage application (*p* > 0.05). *Escherichia coli* O157:H7 and O111 hide prevalence was very low throughout the study; therefore, no further analysis was conducted. However, boot swab monitoring showed a significant reduction in *E. coli* O157:H7, O26, and O45 in the pen floor environment (*p* < 0.05). While using Finalyse as a pre-harvest intervention in the lairage areas of commercial beef processing facilities, this bacteriophage failed to reduce *E. coli* O157:H7 on the hides of beef cattle, as prevalence was low; however, some STECs were reduced in the lairage environment, where the bacteriophage was applied. Overall, an absolute conclusion was not formed on the effectiveness of Finalyse and its ability to reduce *E. coli* O157:H7 on the hides of beef cattle, as prevalence on the hides was low.

## 1. Introduction

In 2021, there were 27.9 million pounds of beef produced, and beef consumption was over 58 pounds per capita, making it the second most consumed meat in the United States [1,2]. However, the beef we eat remains under constant threat of contamination by microbial pathogens, which can subsequently be transmitted to the consumer. Cattle have a very complex natural gut microflora that contains multiple types of bacteria, including pathogens such as Shiga toxin-producing *Escherichia coli* (O157:H7 and six non-O157), which is commonly transmitted by either direct or indirect fecal-oral exposure [3]. *Escherichia coli* O157:H7 has been recognized as a foodborne pathogen since 1982 and is attributed to causing an estimated 73,000 illnesses annually in the U.S. [4]. It is known for its low infectious dose, fewer than 100 cells, and for its ability to produce Shiga toxins (Stx), which are responsible for causing illness in humans. Most cases of *E. coli* O157:H7 initiate with non-bloody diarrhea and self-resolve; however, some progress into bloody diarrhea, known as hemorrhagic colitis, within 1 to 3 days, and 5 to 10% of those cases progress into a life-threatening sequelae disease known as hemolytic uremic syndrome (HUS) or thrombocytopenic purpura (TTP) [5].

In 1994, after a large *E. coli* O157:H7 outbreak, resulting in the death of numerous children from the consumption of undercooked hamburger patties, the U.S. Department of Agriculture Food—Safety and Inspection Service (USDA-FSIS) declared *E. coli* O157:H7 along with six non-O157 Shiga toxin-producing *E. coli* (STEC) adulterants in raw ground beef, non-intact beef cuts, and cuts destined for non-intact beef products [6,7]. This mandated a beef verification testing program for this pathogen and states that any raw products that are contaminated with *E. coli* O157:H7 and/or any of the six non-O157 STECs is unwholesome and cannot be sent into commerce and, therefore, should be rendered unless further processed so the pathogen is destroyed [7]. There are several approaches taken by the beef industry that are used to help improve the safety of beef products in the United States. Most food safety efforts in the beef industry occur post-harvest, which originate on the harvesting floor and consist of antimicrobial interventions, hot water washes, steam pasteurization, proper sanitation/dressing procedures, and good hygiene practices [8]. However, pre-harvest is the origin of pathogen colonization in cattle; therefore, these issues also warrant pre-harvest mitigation strategies geared towards controlling incoming pathogen loads entering commercial abattoirs on harvest-ready feedlot cattle.

Pre-harvest interventions within the beef industry can include the use of direct-fed microbials, bacteriophages, and vaccines [9]. While pre-harvest interventions are not widely adopted in the beef industry, the parallel application of one or more pre-harvest strategies has the potential to synergistically reduce the incidence of human foodborne illness by implementing multiple hurdles against the entry of pathogens into the food chain [8,10]. Bacteriophages (phages) are viruses that infect and replicate only in bacterial cells and kill bacteria. Phages are being used in the food industry as a biocontrol to treat livestock to reduce the fecal shedding of bacterial pathogens, to decontaminate food and non-food contact surfaces, and as a direct application on foods post-harvest [11]. They are the most abundant biological agent on earth, and it is estimated that there are between 10^30^ and 10^32^ phages in the environment [12]. Phages are environmentally safe, self-replicating, and have no deleterious impact on animal and human health or organoleptic properties on the foods to which they are applied [11]. Today, known phages exist to target foodborne pathogens such as *Listeria monocytogenes*, *Salmonella*, *Escherichia coli*, and *Campylobacter jejuni* [11]. However, because phages have a high natural specificity, if the target item/product happens to be contaminated with two or more foodborne bacterial pathogens, a phage preparation targeted against a single pathogen will not be effective in removing non-targeted pathogenic bacteria from foods [13].

Pre-harvest research has demonstrated that the transportation of cattle from the feedlot and holding period in the lairage area of a beef processing facility contributes to a significant increase in *E. coli* O157:H7 hide contamination [14,15]. However, an increase in pathogen prevalence on cattle hides prior to harvest increases the likelihood of carcass contamination on the harvesting floor, which also increases the probability of contamination in the final beef product. Therefore, pre-harvest interventions applied to the live animal at the processing plant would be a valuable addition to the multi-hurdle antimicrobial intervention strategies implemented by commercial beef processing facilities [14,15]. Arm & Hammer™ sells a novel pre-harvest antimicrobial hide wash, Finalyse™ (developed by OmniLytics™—The Phage Company), which is approved by the U.S. Food and Drug Administration (FDA) to be applied, prior to harvest, on beef cattle hides to reduce *E. coli* O157:H7 pathogens. Finalyse™ uses naturally occurring phages to weaken the *E. coli* cell wall and replicate, destroying additional O157:H7 bacteria on the cattle hides [16]. This product is recommended to be applied with their overhead spray system in the cattle lairage area as a spray/shower application on the live cattle prior to the animal entering the processing facility.

A study conducted by Arm and Hammer reported that Finalyse™ effectively reduces *E. coli* O157:H7 by >1.8 log10 CFU/g at an exposure time of five minutes with hide patches that were inoculated with 10^5^ CFU/mL *E. coli* surrogate [17]. Another study, conducted by Arthur et al., applied Finalyse to live cattle in the lairage area of a processing facility and observed an increase of 35.6% and 25.4% in *E. coli* O157:H7 hide prevalence from before to after the phage application for the control and treatment groups, respectively [14]. Overall, an increase in *E. coli* O157:H7 hide prevalence was observed, which is believed to be attributed to the potential for the cattle to acquire additional contamination on their hides as they pass through common spaces and fecal material is transferred to their hides from splashing and pressing up against contaminated surfaces [14]. While it seems that Finalyse might be an effective pre-harvest intervention and has the capability of reducing *E. coli* O157:H7 pathogen prevalence on cattle hides, studies conducted in the lab using hide patches do not account for additional contamination sources that may be present in a lairage environment after the bacteriophage is applied.

A recently proposed ‘regulatory framework’ for controlling *Salmonella* by the FSIS, targeting poultry production, indicated that there is interest in monitoring the pathogen status of the incoming birds at the processing facilities [18]. This framework could eventually be implemented in the lairage area of beef processing facilities for controlling STECs and *Salmonella* as this is the first point FSIS has regulatory control of the live animal and would be able to implement/mandate pre-harvest control measures at this point [19]. Research using Finalyse, or bacteriophages in general, as a pre-harvest intervention on the hides of live cattle in a commercial abattoir is very limited; therefore, further research is needed to determine the efficacy of this intervention in a commercial application. The objective of this study was to evaluate the efficacy of Finalyse, as a pre-harvest intervention, on the reduction in pathogens, specifically *E. coli* O157:H7, on cattle hides and in the lairage environment to overall reduce incoming pathogen loads in the lairage area prior to harvest.

## 2. Materials and Methods

### 2.1. Beef Facility Lairage Area and Hide Application

This study was conducted at a USDA-FSIS-inspected commercial beef harvesting/processing facility located in Eastern Nebraska, where approximately 1800 to 2000 head of beef cattle are harvested/processed daily. This study took place in both the outside holding pens (shown in Figure A1) and the basement holding pens (shown in Figure A2) of the lairage area. The Finalyse intervention was applied to the cattle hides at the end of the basement alley in the lairage area. A total of 30 beef cattle at a time were brought from the outside pens into the basement alley, further referred to as the intervention pen, where the bacteriophage intervention was applied to the group of cattle in a shower application, which hung from the basement ceiling.

Finalyse is a formulation of three naturally occurring phage strains proven to specifically target *E. coli* O157:H7 cells and is used as a pre-harvest hide wash used to aid in the reduction in this pathogen on beef cattle hides prior to cattle entering the harvesting facility [16]. Daily start-up procedures at the facility to ensure the Finalyse solution was properly made and applied include testing water quality to ensure it was ≤50 parts per million (PPM), measuring out 650 mL of Finalyse solution into a carboy. Then, 4.5 gallons of water was added to the same tank (this dilution was made for 1800 head of cattle and was adjusted if more or fewer cattle were being harvested that day), the valve at the bottom of the carboy tank was opened, and the system was then ready to use. The Finalyse was applied at 10^7^ or a dilution ratio of 1:10,000. The application was made to be applied for one minute to 30 head of cattle at a time, and each time the spray system was turned on, 264 ± 10 mL was released from the spray nozzles above the cattle, which is approximately 8.8 mL of Finalyse solution being applied to the hides of each animal.

### 2.2. Hide Swab Collection

Hide swab samples were collected over a two-month period, from June to July 2022, where a total of 300 composite hide swabs (*n* = 150/sampling point) were collected over 5 different sampling events. Hide samples were collected by swabbing approximately a 500 cm^2^ area on the hides of each animal using EZ-Reach 25 mL pre-hydrated buffered peptone water (BPW) sponges (World Bioproducts; Libertyville, IL, USA). It is important to note that a buffering solution was required to inactivate the phage so that it did not continue to kill the *E. coli* cells after the sample was taken; therefore, these pre-moistened swabs were used per the phage companies’ recommendation.

Samples were taken before and after the Finalyse intervention was applied to the hides. Before, hide swabs were taken on the rear of the live cattle, prior to the application of the intervention, as they were entering the intervention pen in the basement area, as outlined in Figure A2. The intervention was then applied to the hides of the cattle, for 1 minute, and it was approximately another 8 to 30 min until the cattle reached the harvesting floor, where the after swabs were taken. The after hide swabs were taken at the beginning of the harvesting floor, at the first legger station, in the same area on the hide as the before swabs. However, no attempt was made to sample the same animals from which the before and after swabs were taken. Samples were taken three times a day (beginning, middle, and end), where a total of 20 composite hide samples were taken at each sampling event; 20 samples were taken at each timepoint: 10 before intervention and 10 after intervention. It is important to note that 1 composite hide swab sample is representative of 5 beef animals; therefore, a total of 1500 different cattle were swabbed during the study period. Upon collection, all samples were immediately chilled and shipped overnight to the International Center for Food Industry Excellence (ICFIE) Food Microbiology Laboratory at Texas Tech University.

### 2.3. Boot Swab Collection

Boot swab samples were taken on the same sampling dates as the hide swab samples, three times a week, from June to July 2022. Fecal material from the pen floor was collected using non-woven shoe covers (VWR; Randor, PA, USA), which will further be referred to as boot swabs. To collect the samples, an individual boot swab was placed over each shoe. The sample was taken by walking from the front to the back of the pen at a normal pace, in a pre-determined ‘Z’ pattern, as shown in Figure A1 and Figure A2. Each boot swab was then removed from the shoe, while wearing gloves, and individually placed in a 1.63 L filtered Whirl-Pak bag (NASCO; Fort Atkinson, WI, USA). A single boot swab was equivalent to one sample. All bags were labeled with the sample number, date, and pen number at the time the sample was taken. Samples were taken from three different locations in the lairage area as follows: (i) Outside pens—before treatment, (ii) Intervention pen—pen where cattle are treated with the Finalyse, and (iii) Inside pens—pens cattle are moved to after being treated, prior to entering the harvesting floor, as displayed in Figure A1 and Figure A2. A total of 171 boot swab samples were collected over the study, where 3–5 boot swab samples were collected/sampling location/sampling event. Samples were immediately chilled and shipped overnight to the International Center for Food Industry Excellence (ICFIE) Food Microbiology Laboratory at Texas Tech University.

### 2.4. Hide Swab and Boot Swab Processing

Upon arrival at the ICFIE Food Microbiological Laboratory, the hide swabs were homogenized in the stomacher (Model 44 circulator, Seward, West Sussex, United Kingdom) at 230 rpm for 1 min. A 1 mL aliquot from each sample was aseptically transferred into a 9 mL tube of Modified Tryptone Soy Broth (mTSB) (Neogen; Lansing, MI, USA). Tubes inoculated with the sample were incubated at 42 °C for 18–24 h. Once the required enrichment time was reached, samples were run on the BAX^®^ Q7 machine using the Real-Time (RT) assay for *Salmonella*, *E. coli* O157:H7 Exact, and STEC—O26, O45, O103, O111, O121, and O145 (STEC Panel 1 and 2).

For each boot swab sample, 100 mL Buffered Peptone Water (BPW) was aseptically added to the 1.63 L sample bag. The sample bags were then homogenized at 230 rpm for 1 min, which will further be referred to as the boot swab homogenate. A 30 mL aliquot of the boot swab homogenate was transferred to a 207 mL filtered Whirl-Pak bag (NASCO; Fort Atkinson, WI, USA). Then, 30 mL of pre-warmed (42 °C) BAX MP media (Hygiena; Camarillo, CA, USA) and 1.0 mL/L Quant™ Solution was added to the 207 mL bag, which will be referred to as the secondary enrichment. The secondary enrichment was incubated at 42 °C for 18–24 h.

### 2.5. Microbial Analysis

Once the required enrichment time was reached, samples were run on the BAX^®^ Q7 machine using the Real-Time (RT) assay for *Salmonella*, *E. coli* O157:H7, and STEC—O26, O45, O103, O111, O121, and O145 (STEC Panel 1 & 2). For the detection of *Salmonella*, *E.* coli O157:H7, and 6 STECs manufacturing guidelines were followed to prepare the sample and hydrate the PCR pellets [20,21]. After hydration, all tubes were placed in the BAX^®^ Q7 instrument, and the full process for BAX^®^ System RT PCR Assay and the appropriate panels were run. At the conclusion of the run, presence/absence (+/−) results were provided at 1 CFU/Sample.

### 2.6. Data Analysis

All data were analyzed using R (Version 4.1.1) statistical software to evaluate the prevalence of pathogens using boot swabs and hide swabs in the lairage area of a commercial beef processing facility to determine the efficacy of a bacteriophage intervention. Prevalence data for each pathogen were evaluated by separate generalized linear models (GLM) with binomial distribution and logit link (logistic regression), and pairwise contrasts were generated by the Tukey method. The results of the analysis provided the odds ratio (OR) with 95% confidence intervals. Confidence intervals were calculated by the ’propCI’ function in the prevalence package using R. Additionally, Fisher’s Exact test was used to determine significance, and all significant differences were evaluated using *p*-values lower than 0.05.

## 3. Results and Discussion

In this study, it is important to note that Finalyse was specifically developed to reduce *E. coli* O157:H7 on the hides of beef cattle; therefore, only the prevalence of *E. coli* O157:H7 was used to determine the efficacy of this bacteriophage. Hide swab and boot swab samples were also evaluated for the presence of *Salmonella* and six other non-O157 Shiga toxin-producing *E. coli* (STEC); however, the evaluation of these pathogens was strictly to measure the changes in pathogen prevalence on the hides and in the pen environment throughout the different locations sampled in the lairage area and to determine if Finalyse had any impact on the other STECs. The results indicated that there was no significant reduction observed for *Salmonella* and/or any Shiga toxin-producing *E. coli* (STEC) on the hides after the bacteriophage application was applied (*p* > 0.05). *Escherichia coli* O157:H7 and O111 hide prevalence was very low throughout the study; therefore, no further analysis was conducted. However, boot swab monitoring in the lairage area of the cattle holding pens showed a significant reduction for *E. coli* O157:H7, O26, and O45 in the pen floor environment (*p* < 0.05). Overall, the data did not determine the efficacy of Finalyse as a pre-harvest intervention in the lairage area of a commercial beef processing facility due to a low *E. coli* O157:H7 prevalence on the cattle hides throughout the study. However, this pathogen was reduced in the pen floor environment where the bacteriophage was being applied. Additionally, this study showed a significant increase in hide contamination for *Salmonella* and STECs as the cattle progressed through the lairage area and acquired additional contamination from the lairage area environment prior to harvest.

### 3.1. Hide Swab Prevalence

*E. coli* O157:H7 prevalence was low on the cattle hides throughout the study period, as shown in Figure 1, and overall prevalence for this pathogen was 0.671% (1/149) and 2.00% (3/150) for before and after hide swabs, respectively. Due to a low prevalence, no further statistical analysis was conducted, and the overall effectiveness of Finalyse as a bacteriophage hide application was not determined through hide swab sampling. While there was an observed increase in prevalence from the before to after hide swabs, it was not significant (*p* > 0.05). An absolute conclusion was not determined, due to the low O157 prevalence, on the efficacy of this bacteriophage with the hide data available; however, the Finalyse intervention did have a measured impact on *E. coli* O157:H7 prevalence in the lairage pens, which is further discussed in Section 3.2.

The overall goal of using this bacteriophage intervention was to reduce *E. coli* O157:H7 hide contamination, which, if successful, would reduce the probability of carcass contamination and in turn reduce/eliminate this pathogen in the final product. Since *E. coli* O157:H7 is considered an adulterant in the industry, all beef trimmings are subject to N60 testing, which is a verification program to help ensure that this pathogen does not remain in the final raw ground beef product [6,7]. Reducing the occurrence of *E. coli* O157:H7 positives in beef trimmings would allow for the facility to save money on the additional labor, packaging, and storage costs they incur when re-boxing products and a potential recall and public shaming if adulterate products entered into commerce. Additionally, this contaminated product must be sent for further processing into a fully cooked product instead of remaining fresh for ground beef, which results in a decreased value of the product.

For this specific facility, using the bacteriophage pre-harvest intervention in the lairage environment cost USD 0.60/head (based on 1800 head/day), which was a more desirable upfront cost to controlling this pathogen in the lairage area than having to control it at the end of the process in the final product. While the Nebraska facility did mention that they observed a decrease in the occurrence of event days over the summer months, when the bacteriophage intervention was being applied, it is not believed to be attributed to the application of the Finalyse intervention, as *E. coli* O157:H7 was not prevalent on the hides in the first place. As mentioned, the data reveal that *E. coli* O157:H7 hide contamination was low throughout the study prior to the intervention being used; therefore, if there is minimal contamination on the hides pre-harvest, then there is also minimal probability for carcass contamination to occur on the hides in the lairage area and in the carcasses on the harvesting floor. Overall, it would be prejudiced to attribute a reduction in *E. coli* O157:H7 event days to the use of this bacteriophage intervention without the support of the hide data.

A study conducted by Arm and Hammer reported that Finalyse effectively reduces *E. coli* O157:H7 by >1.8 log10 CFU/g at an exposure time of five minutes with hide patches that were inoculated with 10^5^ surrogate CFU/mL [17]. However, this study does not account for hide contamination acquired in the lairage area of beef processing facilities. Furthermore, a study conducted by Arthur et al. applied Finalyse™ to live cattle in the lairage area of a beef processing facility and observed an increase in *E. coli* O157:H7 hide prevalence from before to after the phage application [14]. Before hide swabs were taken, the cattle were unloaded in the lairage area of the processing facility, and *E. coli* O157:H7 prevalence was 6.4% and 13.3% for the control and the treated groups, respectively [14]. The phage application was applied to the hides of the treatment groups, and the after hide swabs were taken once the cattle were shackled on the harvesting floor, at least one hour after the phage application. After the phage application, *E. coli* O157:H7 prevalence on the hides was 42% and 38.7% for the control and the treated groups, respectively. Overall, an increase in *E. coli* O157:H7 hide prevalence was observed, which is believed to be attributed to the potential for the cattle to acquire additional contamination of their hides as they pass through the common spaces and fecal material is transferred to their hides from splashing and pressing up against a contaminated surface [14]. This was also a trend that was commonly observed for the other pathogens tested in this study—*Salmonella* and five of the six non-O157 STECs, which are discussed in the following paragraphs of Section 3.1.

Since bacteriophages are pathogen-specific, Finalyse is not a pre-harvest intervention used to target the reduction in other pathogens, such as *Salmonella* and six non-O157 STEC, which are commonly shed by cattle. The prevalence of these pathogens on the cattle hides was evaluated to determine if Finalyse had an impact on any of the other STECs or if any trends regarding cross-contamination on the hides occurred within the lairage area. For these pathogens mentioned, there was a significant increase in prevalence from before to after the intervention was applied (*p* < 0.05), and it was more likely for these pathogens to be detected after the intervention than before the intervention when compared pairwise, as shown in Table 1. The average *Salmonella* hide prevalence, as shown in Table A1, was 28.9% (43/149) and 61.3% (92/150) for before and after hide swabs, respectively. From the time the cattle were moved into the basement alley, where the before swabs were taken, to the time the after swabs were taken on the harvest line, there was a 32.4% increase in *Salmonella* hide prevalence. Again, Finalyse was not designed to reduce *Salmonella*, but these data illustrate that significant cross-contamination occurs in the lairage area. The statistical analysis, shown in Table 1, indicated that the increase in *Salmonella* after hide prevalence was significant (*p* < 0.001) and is 3.96 times more likely to be detected than before the intervention application.

A study conducted by Beach et al. evaluated the prevalence of *Salmonella* on the hides of beef cattle from transport to slaughter and indicated a 38% increase in *Salmonella* prevalence from pre-transit to post-transit, which led to carcass contamination of 19% in feedlot cattle [22]. For adult cattle, there was a 32.4% increase in *Salmonella* prevalence from pre-transit to post-transit, which led to 54.2% of carcasses being contaminated with *Salmonella*. This study indicated that potential sources of *Salmonella* contamination, from pre-transit to slaughter, included transport vehicles, holding pens, knock boxes, workers, and equipment [22]. Additionally, another study, conducted by Brichta-Harhay et al., also evaluated *Salmonella* contamination on the hides and carcasses of cattle and reported a hide prevalence of 86–94% and a carcass prevalence of 44–55% throughout the four different seasons of the year, indicating that *Salmonella* hide prevalence significantly contributed to carcass contamination [23]. While *Salmonella* is not considered an adulterant in the beef industry, it is still a major pathogen of concern and efforts need to be considered to reduce this pathogen prior to harvest. This study reveals that the lairage area at this facility, like other facilities, is a contributor to increased *Salmonella* hide contamination as the cattle progress through the lairage area. It is concerning that a >30% increase in *Salmonella* prevalence was attributed to cross-contamination in the lairage area environment and can reduce the effectiveness of post-harvest interventions if there are high incoming pathogen loads on the cattle.

Lastly, a significant increase in pathogen hide prevalence from before to after hide samples was observed for STEC—O26, O121, O45, O103, and O145, as shown in Figure 1. There was also an increase observed for *E. coli* O111; however, the prevalence was low throughout the study period, 3.36% (5/149) and 4.00% (6/150) for before and after hide swabs, respectively, as shown in Table A1; therefore, no further analysis was conducted. *Escherichia coli* O26 hide prevalence was 80.5% (120/149) and 98.7% (148/150) for before and after hide swabs, respectively, as shown in Table A1. The statistical analysis, shown in Table 1, indicated that it was significantly more likely for this pathogen to be detected after the intervention was applied than before (OR = 35.5, 95% CI: 5.71, 1460, *p* < 0.001), indicating that pathogen contamination from the lairage area increased by 18.2% from the time the before swabs were taken to when after swabs were taken. These results remain the same for the other STECs mentioned below. A significant increase of 18.1% for *E. coli* O121 hide prevalence was observed, as the overall prevalence shown in Table A1 for this pathogen was 79.9% (119/149) and 98.0% (147/150) for before and after hide swabs, respectively (Table 1: OR = 18.4, 95% CI: 4.51, 162, *p* < 0.001), which was similar to the increase in *E. coli* O26. Additionally, overall *E. coli* O45 hide prevalence was 72.5% (108/149) and 95.3% (143/150) for before and after hide swabs, respectively (Table 1: OR = 8.99, 95% CI: 3.62, 26.9, *p* < 0.001). This STEC had the third lowest before prevalence of the six STECs; however, it had a large percent increase in prevalence, 22.8%, from before to after hide swabs. Overall hide prevalence for *E. coli* O103 was 87.2% (130/149) and 98.7% (148/150) for before and after hide swabs, respectively. As shown in Figure 1 and Table A1, this pathogen had the highest prevalence before the intervention was applied than any of the other pathogens and had an increase in prevalence of 11.5% (Table 1: OR = 21.5, 95% CI: 3.32, 901, *p* = 0.003). Lastly, overall, *E. coli* O145 hide prevalence for the study period was 10.1% (15/149) and 20.0% (30/150) for before and after hide swabs, respectively. The prevalence of this pathogen on the cattle hides is relatively lower than the occurrence of the other STECs; however, the 9.9% increase in prevalence from before to after swabs was significant (*p* < 0.05), with an OR equal to 2.34 (95% CI: 1.16, 4.91), as shown in Table 1.

Overall, *Salmonella* had the largest percent increase from before to after swabs, followed by *E. coli*—O25, O26, and O121—indicating that significant hide contamination occurs in the basement area of the lairage. It is also important to note that this contamination was not from the use of Finalyse intervention, but most likely due to cattle rubbing up against each other and contaminated surfaces in the lairage area. This hide contamination contributed by the lairage environment has the potential to contribute to carcass contamination during harvest; therefore, it is important to acknowledge that control measures need to be put in place by processors to mitigate these pathogens and reduce overall cross-contamination in the lairage area environment.

### 3.2. Boot Swab Prevalence

The boot swab data indicated a significant reduction for *E. coli* O157:H7, O26, and O45 between the intervention pen and the outside pen and/or the intervention pen and the inside pens over the course of the study (*p* < 0.05). This implies that the intervention pen, which is where the Finalyse intervention was being applied to the cattle hides, was a low-risk area for potential pathogen contamination, as the odds ratios, shown in Table 2, ranged from 0 to 0.28, implying that Finalyse, which remains in the pen floor environment, may have contributed to the reduction in *E. coli* O157:H7, O26, and O45 in the pen floor environment. The boot swab sampling method was used in this study to monitor the lairage area environment, as it was validated in a preliminary study conducted by Flach et al. and proven to be effective at detecting STECs and *Salmonella* in the pen floor environment [24].

The overall boot swab prevalence for *E. coli* O157:H7 was 35% (20/57), 0.0% (0/57), and 0.0% (0/57) for the outside pens, intervention pen, and inside pens, respectively, as shown in Figure 2 and Table A2. A significant reduction was observed between the intervention pen and the outside pen (Table 2: OR = 0.0; 95% CI: 0, 0.14, *p* < 0.001), and the same significant reduction was also observed between the inside pen and the outside pen (*p* < 0.001). The boot swab data indicate that there was a low probability for O157 contamination to occur on the hides from the lairage area environment and further explain the reason why a significant increase in *E. coli* O157:H7 hide prevalence, as discussed in Section 3.1, was not observed, because O157 prevalence was 0.0% over the course of the study in the intervention pen and the inside pens (the basement lairage area identified in Figure A2). Considering that the outside holding pens were the only area O157 was prevalent in the lairage area, this is where cross-contamination on the hides from the lairage environment would have occurred; however, the overall *E. coli* O157:H7 hide contamination was <1%, as shown in Table A1, for the before hide swabs, indicating that pathogen contamination from the outside pens onto the cattle hides was minimal in this area and did not increase significantly as the cattle progressed through the lairage area. Overall, more data in a commercial setting are needed to determine the efficacy of Finalyse on the reduction in *E. coli* O157:H7; however, in this study, it was effective in eliminating this pathogen from the lairage area environment (in the intervention pen), and no increase in O157 hide prevalence was observed as the cattle progressed through the lairage area.

While Finalyse is not a bacteriophage intervention designed to target other STECs, a reduction in *E. coli* O26 and *E. coli* O45 was also observed in the intervention pen located in the basement of the lairage area. The overall *E. coli* O26 prevalence was 82% (42/51), 42% (23/55), and 72% (39/54) for the outside pens, intervention pen, and inside pens, respectively, as shown in Table A2. Although the reduction from the outside pen to the inside pen was not significant (*p* > 0.05), the reduction from the outside pen to the intervention pen (Table 2: OR = 0.16, 95% CI: 0.06, 0.41, *p* < 0.001) and the inside pen to intervention pen (Table 2: OR = 0.28, 95% CI: 0.11, 0.66, *p* < 0.05) was significant. Once again, these data correlate with and help further explain the increase in *E. coli* O26 hide contamination as the cattle progress through the lairage area. As previously mentioned, the O26 prevalence in the outside lairage pens was high, 82%, and before hide swab prevalence, which was taken as the cattle were brought into the basement area from the outside pens, was comparable at 81%, as shown in Table A1. A 41% prevalence reduction was then observed from the outside pens to the intervention pen, which the Finalyse intervention could have solely contributed to or been one of a variety of factors that contributed to this reduction. However, a 30% prevalence increase was observed from the intervention pen to the inside pens, which are the pens the cattle are held in/moved through after the intervention was applied. The hide swab data indicated an 18% O26 prevalence increase from the before to after hide swabs, indicating cross-contamination occurred from the lairage environment (inside pens), increasing hide contamination pre-harvest. The pen floors in the lairage area were heavily soiled with a slurry of fecal material during sampling, which, identified by the boot swab samples, is contaminated with pathogens, and this contamination can end up on the cattle hides and leads to an increase in prevalence prior to harvest.

Furthermore, the overall *E. coli* O45 prevalence was 90% (46/51), 72% (41/57), and 88% (45/51) for outside pens, intervention pens, and inside pens, respectively, as shown in Table A2. A significant reduction was observed between the outside pen and intervention pen (Table 2: OR = 0.28, 95% CI: 0.18, 3.47, *p* < 0.05), and even though not significant, there was also a reduction observed between the inside pens and the intervention pen (Table 2: OR = 0.34, 95% CI: 0.10, 1.04, *p* = 0.05) as shown in Table 2. Overall, a 16% prevalence reduction for this STEC from the outside pens to the intervention pen was observed, which indicates that the Finalyse intervention may have contributed to *E. coli* O45 in the pen floor environment in the pen floor environment. Similar outcomes were observed in relation to the hide swab data as the pathogen previously discussed. *Escherichia coli* O45 prevalence in the outside pens was 88%, indicating a high-risk area for contamination to occur on the hides if not already previously contaminated during transit. While the boot swab data show that Finalyse may have contributed to the reduction of O45 prevalence in the pen floor environment of the intervention pen, these data also correlate with the 18% increase in O45 prevalence from the intervention pen to the inside pens to the 23% prevalence increase from the before to after swabs, indicating that the inside pen environment also contributed significantly to O45 hide contamination.

Overall, *Salmonella* prevalence for the study period was 100% (57/57), 97% (55/57), and 98% (56/57) for outside pens, intervention pens, and inside pens, respectively. No significant reductions were observed between sampling locations, as the prevalence of this pathogen was relatively high in all areas. It can be concluded that *Salmonella* was a highly prevalent pathogen in the lairage area of this processing facility and should encourage better management strategies in the lairage area to reduce the potential for contamination to occur right before harvest. Additionally, overall, *E. coli* O121 prevalence was 88% (45/51), 89.1% (49/55), and 87% (47/54) for outside pens, intervention pens, and inside pens, respectively. Overall, *E. coli* O103 prevalence was 62.7% (32/51), 47.4% (27/57), and 59% (30/51) for outside pens, intervention pens, and inside pens, respectively. *Escherichia*
*coli* O145 prevalence was 28% (14/51), 26% (15/57), and 16% (8/50) for outside pens, intervention pens, and inside pens, respectively. Lastly, *E. coli* O111 prevalence was low; therefore, no further statistical analysis was conducted. Overall O111 prevalence was 0% (0/51), 0% (0/55), and 1.9% (1/54) for outside pens, intervention pens, and inside pens, respectively, as shown in Table A2. In the preliminary study conducted by Flach et al., which was conducted at the same facility as this study, *E. coli* O111 prevalence was 0% for all three sampling methods, therefore indicating a low potential for presence and cross-contamination in the lairage area [24].

Overall, these data indicate that there was potential throughout every area in the lairage area sampled for the live cattle to acquire additional pathogen contamination. While it was expected that Finalyse was not going to have a reduction potential for any additional STECs, it is important to acknowledge that multi-hurdle approaches (pre-harvest interventions, effective management strategies, effective cleaning/sanitation procedures) will be required to reduce the potential for cross-contamination that occurs in the lairage environment. While producers are responsible for pathogen colonization at the farms/feedlots, producers need to take accountability for the cross contamination on the hides occurring in the lairage prior to harvest. Effective management strategies are warranted to control more than one pathogen in the pen floor environment of the lairage area to reduce incoming pathogen loads entering commercial abattoirs.

## 4. Conclusions

It is apparent that foodborne pathogens, such as Shiga toxin-producing *E. coli* and *Salmonella*, are prevalent in the lairage area of commercial beef processing facilities and contribute to significant cross-contamination on the hides as the cattle progress through the lairage area. The beef industry currently relies on a multi-hurdle approach of post-harvest interventions that take place on the harvest and fabrication floor to reduce/eliminate the incoming pathogen loads on the carcass; however, these approaches are not inevitable to the occurrence of pathogen contamination in the final meat product. The renewed interest in pre-harvest food safety from both a regulatory and public health perspective requires a revived interest in pre-harvest pathogen presence and control, specifically in the lairage areas of a commercial beef processing facility. While our data did not determine the efficacy of Finalyse as a pre-harvest bacteriophage application, it identified data gaps in the research and literature that need to be further investigated in order to determine the efficacy of this bacteriophage, Finalyse. Due to a low *E. coli* O157:H7 prevalence on the cattle hides before and after the intervention, the efficacy of Finalyse as a bacteriophage hide wash was not concluded. However, the boot swab data, monitoring the pen floor environment, did identify a significant reduction in the intervention pen for *E. coli* O157:H7, *E. coli* O26, and O45; nevertheless, it is unsure if the Finalyse intervention is one hundred percent responsible for these reductions, considering Finalyse specifically targets O157:H7. Additionally, for the other STECs and *Salmonella* there was an observed increase in prevalence from the before to after hide swabs (*p* < 0.05), which, indicated by the boot swab results, was due to environmental contamination the cattle acquired as they progressed through the lairage area. Significant increases in hide contamination before the cattle enter the harvest floor warrant mitigation strategies geared towards controlling incoming pathogen loads and efforts to reduce contamination in the lairage area. Overall, further research is needed to evaluate the efficacy of Finalyse as a pre-harvest bacteriophage intervention in a commercial beef processing facility to take into account the additional cross-contamination that can occur after the intervention is applied.

## Figures and Tables

**Figure 1 foods-12-04349-f001:**
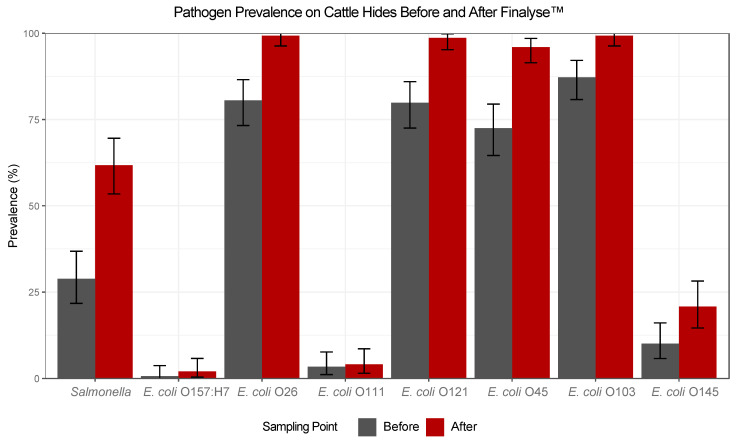
Pathogen prevalence (%) on cattle hides before and after the application of Finalyse in the lairage area of a commercial beef processing facility over the course of two months. A total of 300 composite hide swabs were taken (*n* = 150 swabs/sampling point). Error bars represent a 95% confidence interval (CI).

**Figure 2 foods-12-04349-f002:**
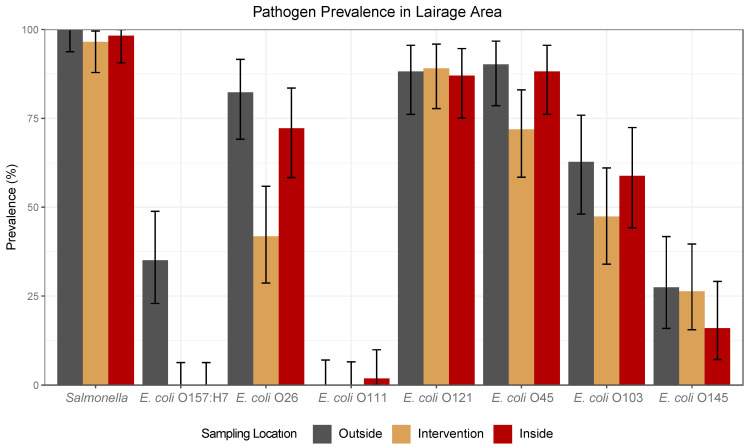
Overall pathogen prevalence (%) on the pen floor environment from different locations (outside pens—before intervention, intervention pen—where the bacteriophage was applied, and inside pens—after the intervention was applied) in the lairage area of a commercial beef processing facility from boot swab samples. A total of 171 boot swab samples were taken over the course of the study period. Error bars represent 95% confidence intervals (CI).

**Table 1 foods-12-04349-t001:** Odds ratios (ORs), 95% confidence intervals (CIs), and probability (*p*) values for the likelihood of detectable pathogens on cattle hides before and after the application of Finalyse™ from GLM analysis.

Pathogen	Contrast	OR	95% CI	*p*-Value
*Salmonella*	After–Before	3.96	2.38–6.66	<0.001
*E. coli* O26	After–Before	35.5	5.71–1460	<0.001
*E. coli* O121	After–Before	18.4	4.51–162	<0.001
*E. coli* O45	After–Before	8.99	3.62–26.9	<0.001
*E. coli* O103	After–Before	21.5	3.32–901	0.003
*E. coli* O145	After–Before	2.34	1.16–4.91	0.012

**Table 2 foods-12-04349-t002:** Odds ratio (OR), 95% confidence intervals (CI), and probability (*p*) values for the likelihood of detectable pathogens in the cattle holding pens at three different sampling points (outside, intervention, and inside) from GLM analysis.

Pathogen	Contrast	OR	95% CI	*p*-Value
*Salmonella*	Intervention–Inside	0.49	0.01–9.75	1.00
	Intervention–Outside	0	0–5.31	0.50
	Outside–Inside	0	0–39.00	1.00
*E. coli* O157:H7	Intervention–Inside	0	0–inf	1.00
	Intervention–Outside	0	0–0.14	<0.001
	Outside–Inside	0	0–0.14	<0.001
*E. coli* O26	Intervention–Inside	0.28	0.11–0.66	<0.05
	Intervention–Outside	0.16	0.06–0.41	<0.001
	Outside–Inside	0.56	0.19–1.55	0.25
*E. coli* O121	Intervention–Inside	1.23	0.32–4.72	0.78
	Intervention–Outside	1.09	0.27–4.40	1.00
	Outside–Inside	0.90	0.23–3.38	1.00
*E. coli* O45	Intervention–Inside	0.34	0.10–1.04	0.05
	Intervention–Outside	0.28	0.07–0.90	0.03
	Outside–Inside	0.82	0.18–3.47	1.00
*E. coli* O103	Intervention–Inside	0.63	0.27–1.44	0.25
	Intervention–Outside	0.54	0.23–1.23	0.12
	Outside–Inside	0.85	0.35–2.03	0.84
*E. coli* O145	Intervention–Inside	1.86	0.66–5.65	0.24
	Intervention–Outside	0.94	0.37–2.42	1.00
	Outside–Inside	0.51	0.16–1.48	0.23

## Data Availability

Data are contained within the article.

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
