# Peer review of "Validation of a Bacteriophage Hide Application to Reduce STEC in the Lairage Area of Commercial Beef Cattle Operations"

_foods, 2023, doi:10.3390/foods12234349_

Round 1

Reviewer 1 Report

Comments and Suggestions for Authors

The manuscript entitled "Validation of a Bacteriophage Hide Application to Reduce STEC in the Lairage area of Commercial Beef Cattle Operations" highlights the use of bacteriophage as antibacterial therapy to reduce the microbial load particulary E. coli from animals to decrease the meat contamination. 

Although, the article is well written but the major concern is the suitability and significance of the study in wider use in industry etc. 

In my opinion, I suggest the authors to discuss why this study is important for scineitific community and how this study can benefit the industry?

Comments on the Quality of English Language

The English language is appropriate, minor grammatical and spell check required. 

Author Response

Dear Editor and Reviewers,

We would like to thank reviewers for their comments regarding our manuscript titled: “Validation of a Bacteriophage Hide Application to Reduce STEC in the Lairage area of Commercial Beef Cattle Operations”. We revised our manuscript to address the comments as explained in detail in this response.  

Please consider this revised manuscript towards our submission of this article in Foods. 

Kind regards, 

The Authors 

Reviewer 1 -

  1. Although, the article is well written, but the major concern is the suitability and significance of the study in wider use in industry etc.
    1. Added an addition statement in the introduction, to help establish significance, regarding the potential interest of the FSIS monitoring the incoming status of live cattle at the abattoirs and require processing facilities to control pathogens such as STECs and Salmonella prior to harvest (please see lines 116-122). Since cattle producers are not regulated by the FSIS they cannot be mandated to control pathogens pre-harvest at the feedlots to reduce incoming pathogen loads at the abattoirs, therefore, leaving cattle producers responsible for controlling pathogen loads on the cattle in the lairage area. Additionally, beef processors are being held accountable for the human foodborne illness rate related to beef and are targeting a 15% reduction by 2023 (Healthy people 2023 goals). Beef processors have made significant advances over the years to reduce pathogens post-harvest and while a multi-hurdle post-harvest intervention approach has proven to be effective in reducing foodborne pathogens in the final meat product, it does not guarantee elimination of pathogens from the final product. Therefore, recent proposed research efforts have been geared towards reducing pathogens in the live animal pre-harvest to help alleviate the dependence on post-harvest interventions once the pathogen is already there contaminating the beef product.
    2. Bacteriophages are not a new novel technology; however, they are becoming more demanding in the lairage area setting of beef processing facilities to help reduce the incoming pathogen loads pre-harvest. Finalyse, the bacteriophage used in this study has proven to be an effective pre-harvest intervention, in a laboratory setting, to reduce an E. coli O157:H7 cocktail on hide patches. However, this reduction has not been replicated in a commercial setting based on the literature reviewed. Therefore, it is important that pre-harvest interventions are implemented and studied in commercial settings to prove effectiveness on the live animal and consider other factors that occur in the lairage area.
    3. In addition to monitoring the prevalence of O157:H7 on the hides we also monitored 6 non-O157:H7 STECs and Salmonella to determine prevalence on the hides and in the pen floor environment to determine how it changed as the cattle progressed through the lairage area. The authors believe that it was beneficial to take boot swab samples in addition to the hide swab samples as the lairage area environment was able to help explain the increase in pathogen prevalence on the hides as the cattle progressed through the lairage area. Since the O157 prevalence on the hides was low, the effectiveness of Finalyse as a pre-harvest intervention was not determined. However, it is still important to share the research and encourage the replication of this study in a commercial setting when E. coli O157:H7 prevalence is high on the hides to truly determine the effectiveness. Overall, the authors believe that this research is suitable and relevant to the beef industry and controlling pathogens pre-harvest. This has been emphasized in the introduction, discussion, and conclusion. Please reference these sections.
  1. In my opinion, I suggest the authors to discuss why this study is important for scientific community and how this study can benefit the industry?
    1. Agree, this has been emphasized on in the conclusion section of the article. (Please see lines 504 -530 in the tracked changes version)

Reviewer 2 Report

Comments and Suggestions for Authors

The manuscript needs revision. Please refer to comments given in the text of reviewed attached file of the manuscript.

Author Response

Dear Editor and Reviewers,

We would like to thank reviewers for their comments regarding our manuscript titled: “Validation of a Bacteriophage Hide Application to Reduce STEC in the Lairage area of Commercial Beef Cattle Operations”. We revised our manuscript to address the comments as explained in detail in this response.  

Please consider this revised manuscript towards our submission of this article in Foods. 

Kind regards, 

The Authors 

Reviewer 2 –

  1. What is the superiority of your research compared to other researchers?
    1. In the introduction, it mentions that this study considers the additional contamination that may occur in the lairage area after the bacteriophage is applied to the cattle hides. This is one thing that studies conducted in a laboratory setting using hide patched does not consider. This study is conducted in a commercial application and the bacteriophage was applied in a set up per the phage companies’ recommendation. While this study is similar to other hide prevalence research in commercial settings, this study uses boot swabs in addition to hide swabs to monitor the lairage area environment While we were not able to determine the effectiveness of the Finalyse intervention, this study can be replicated at a different commercial beef processing facility and/or during a season when O157 prevalence is high on the hides. Additionally, this study continues to bring to light that the lairage area leads to significant cross contamination on the hides of beef cattle as they continue to progress through the lairage area indicating that they are increasing the pathogen loads on the cattle prior to harvest increasing the chances of cross contamination on the slaughter floor to the carcass. (Please reference the introduction section.) This has also been expanded on in the discussion and conclusion sections.
  2. Please add your conclusion at the end of the abstract section
    1. Conclusion has been added to the end of the abstract section. (Please see lines 28-30 in the tracked changes version)
  1. In the text of the introduction, it is not clear what the problem is? And what problem do you want to solve?
    1. The problem is that cattle harbor and shed pathogens, such as Shiga toxin-producing E. coli (STEC) and Salmonella, which contaminates the hides and during harvest can contaminate the carcass and end up in the final meat product causing illness to humans. Most research efforts in the past have focused on how to control these pathogens post-harvest on the meat and while they can be effective, they do not completely eliminate the chance of pathogen contamination in the final meat product. However, pre-harvest is the origin of pathogen colonization; therefore, efforts are needed to reduce the incoming pathogen loads on cattle prior to harvest, which can aide in pathogen reduction post-harvest. The authors added a statement addressing the potential interest for the FSIS to monitor the incoming pathogen loads on live cattle in the lairage area prior to harvest (please see lines 114-119 in the tracked changes version). It is important that processors get ahead of the government agency and determine effective strategies that can be implemented in the lairage area to control pathogens prior to harvest.
    2. The purpose of this study was to determine if Finalyse is effective at reducing E. coli O157:H7 on cattle hides in a commercial setting. However, research determining the effectiveness of this bacteriophage is limited and is proven to be effective in a laboratory setting. Laboratory settings do not consider the additional contamination that can occur in the lairage area after the intervention was applied. The problem and the problem the authors want to solve is addressed in the introduction.
      1. Main problem: Research determining the effectiveness of Finalyse as a pre-harvest intervention reducing E. coli O157:H7 on cattle hides in a commercial setting is limited.
      2. Problem want to solve: Determine the effectiveness of Finalyse on the reduction of E. coli O157:H7 on the hides of cattle prior to harvest in a commercial setting. Additionally, we want to determine how contamination in the lairage environment contributes to changes in hide contamination.
  1. Please specify in the objective whether your research is being for the first time in the world or is it a continuation of another research?
    1. The authors disagree that this statement needs to be included in the objective statement. It can be identified from the introduction that this is not the first-time research like this has been conducted. Finalyse is not a new and novel bacteriophage intervention. Finalyse has been on the market and used in the industry for some time now and has proved to be effective in a laboratory setting. However, there is minimal data on the effectiveness of this bacteriophage in a commercial setting. The authors identified that there is a need for further research of Finalyse in a commercial application. This is already addressed in the introduction.
  1. It is better to explain about importance of meat production and its application. For this you can use below added sentences and references: Moreover, farm-animal species play crucial roles in satisfying demands for meat on a global scale, and they are genetically being developed to enhance the efficiency of meat production (Mohammadabadi et al., 2010). In particular, one of the important breeders’ aims is to increase skeletal muscle growth in farm animals (Shokri et al., 2023). The enhancement of muscle development and growth is crucial to meet consumers’ demands regarding meat quality (Jafari Ahmadabadi et al., 2023).

Jafari Ahmadabadi SAA, Askari-Hemmat H, Mohammadabadi M, Asadi Fouzi M, Mansouri M 2023. The effect of Cannabis seed on DLK1 gene expression in heart tissue of Kermani lambs. Agricultural Biotechnology Journal 15 (1), 217-234.

Mohammadabadi MR, Torabi A, Tahmourespoor M, Baghizadeh A, 2010. Analysis of bovine growth hormone gene polymorphism of local and Holstein cattle breeds in Kerman province of Iran using polymerase chain reaction restriction fragment length. African Journal of Biotechnology 9 (41), 6848-6852.

Shokri S, Khezri A, Mohammadabadi M, Kheyrodin H 2023. The expression of MYH7 gene in femur, humeral muscle and back muscle tissues of fattening lambs of the Kermani breed. Agricultural Biotechnology Journal 15 (2), 217-236.

    1. The authors disagree with this statement as these sentences provided above by the reviewer are not relevant to the aim of this study or applicable to the introduction. The importance of meat production is briefly discussed in the first sentence as beef is the second most consumed meat in the U.S. (please see lines 35-36).

Reviewer 3 Report

Comments and Suggestions for Authors

 The research design is good. However, the English is "ponderous". Trying so hard to be scientific that it becomes incomprehensive. It would be helpful if the comments were linked to the tables and figures more closely. Simplification is required.

Also, the argument that a specific phage meant for 0157 seems to decrease levels of  Salmonella is fairly novel and needs more emphasis  and explanation. WHY??

Comments on the Quality of English Language

See above. Please  simplify the language and link comments to relevant tables and figures. Also the finding that a bacteriophage meant to remove E coli 0157 removes Salmonella needs more explanation. The fact  that there was no significant difference of E coli 0157 on the hides  is interesting. I suspect that the finding in regard to "boots" is very significant as it may be that the water on the floor, contaminated by ingesta, is the real culprit for 0157 in meat as that soiled water is often splashed up and possibly tiny droplets in the air are contaminating the carcasses. I have always suspected that the "hide contamination" could not be the main reason for 0157 in meat. Droplet contamination seems feasible and  thy could suggest that this be researched further, in their conclusions.

Author Response

Dear Editor and Reviewers,

We would like to thank reviewers for their comments regarding our manuscript titled: “Validation of a Bacteriophage Hide Application to Reduce STEC in the Lairage area of Commercial Beef Cattle Operations”. We revised our manuscript to address the comments as explained in detail in this response.  

Please consider this revised manuscript towards our submission of this article in Foods. 

Kind regards, 

The Authors 

Reviewer 3 –

  1. The research design is good. However, the English is "ponderous". Trying so hard to be scientific that it becomes incomprehensive. It would be helpful if the comments were linked to the tables and figures more closely. Simplification is required.
    1. The results and discussion section were reviewed and major revisions were made to make the results and discussion of the article more comprehensive. Ponderous language was removed from the article. Additionally, comments in the results and discussion section have been linked to the tables and figures in the results and discussion section. (Please reference the results and discussion section).

  1. Also, the argument that a specific phage meant for 0157 seems to decrease levels of  Salmonella is fairly novel and needs more emphasis and explanation. WHY??
    1. Finalyse is a bacteriophage that specifically targets E. coli O157:H7. As mentioned in the introduction, phages have a high natural specificity and are not effective in removing non-targeted pathogenic bacteria. This study was not targeting the reduction of Salmonella; however, we wanted to determine Salmonella Prevalence on the hides and in the pen floor environment to determine how the lairage environment impacts pathogen cross-contamination to the hides. The results do not mention a reduction of Salmonella in the hides or in the pen floor environment, as there was an increase in Salmonella from the before to after hide swabs and greater than 95% overall prevalence in the pen floor environment for all three sampling areas. Finalyse application in the lairage area had no impact on the reduction of Salmonella in this study and further proves that bacteriophages are pathogen specific and the lairage area can contribute to further hide contamination prior to harvest. Please review the results section and figure 1 and 2 for any further explanation regarding the prevalence of Salmonella.
  1. Comments on the Quality of English Language

See above. Please  simplify the language and link comments to relevant tables and figures. Also the finding that a bacteriophage meant to remove E coli 0157 removes Salmonella needs more explanation. The fact  that there was no significant difference of E coli 0157 on the hides  is interesting. I suspect that the finding in regard to "boots" is very significant as it may be that the water on the floor, contaminated by ingesta, is the real culprit for 0157 in meat as that soiled water is often splashed up and possibly tiny droplets in the air are contaminating the carcasses. I have always suspected that the "hide contamination" could not be the main reason for 0157 in meat. Droplet contamination seems feasible and  thy could suggest that this be researched further, in their conclusions.

    1. The results and discussion section was reviewed and the English was simplified (please refer to the results and discussion section). As previously mentioned, this study did not conclude that Finalyse was effective at reducing Salmonella on the hides of beef cattle, I think there was a misunderstanding here. The authors agree that it was interesting to find E. coli O157 in the environment but not on the hides, which is why further research needs to be conducted before a conclusion on the efficacy of Finalyse can be determined. It is hard to 100% attribute the reduction of O157 in the basement area to the Finalyse intervention as there may be other factors here like you mentioned and the data would not 100% support that conclusion.

Reviewer 4 Report

Comments and Suggestions for Authors

The manuscript foods-2621318  presents the efficacy of Finalyse, as a pre-harvest intervention, on the reduction of pathogens on the cattle hides to overall reduce incoming pathogen loads.

The subject is of great interest and the authors provide enough details to support the aim of the study. 

The authors tested STEC and Salmonella spp. In the title only STEC is mentioned. Also, what Salmonella spp. was tested? 

The limitations of the study are not clearly mentioned. 

minor comments. Please correct in italics all the latin names of bacteria e.g L202 Salmonella, E.coli, etc.

keywords should reflect the research that was conduced. Why only E.coli O157:H7?

Author Response

Dear Editor and Reviewers,

We would like to thank reviewers for their comments regarding our manuscript titled: “Validation of a Bacteriophage Hide Application to Reduce STEC in the Lairage area of Commercial Beef Cattle Operations”. We revised our manuscript to address the comments as explained in detail in this response.  

Please consider this revised manuscript towards our submission of this article in Foods. 

Kind regards, 

The Authors 

Reviewer 4:

  1. The authors tested STEC and Salmonella spp. In the title only STEC is mentioned. Also, what Salmonella spp. was tested?
    1. Shiga toxin-producing E. coli was the main focus in this study, more specifically E. coli O157:H7, as this is what the bacteriophage Finalyse specifically targets. As discussed in the article, bacteriophages are very pathogen specific and are not effective in destroying non-targeted pathogenic bacteria. The authors did not want to confuse the readers by focusing on the reduction of Salmonella on cattle hides or in the pen floor environment as it was not likely this would occur and after seeing the results did not occur. We still wanted to include Salmonella in the analysis to evaluate its prevalence in the lairage area and how it impacts hide contamination in the lairage area.
    2. No specific species of Salmonella was tested for. The BAX Real-Time Salmonella PCR assay was used to determine Salmonella prevalence for both hide swab and boot swab samples. This test can identify 49 strains of Salmonella from 39 serotypes. (Source - https://www.food-safety.com/articles/5982-baxc2ae-system-real-time-pcr-assay-for-salmonella-excels-in-challenging-food-matrices).
  1. The limitations of the study are not clearly mentioned.
    1. Limitations regarding the microbial analysis are reported in the methodology section and is 1 CFU/sample (please see line 218-219). Limitations regarding the experimental design have been expanded on in the discussion of the article, specifically in the boot swab section.
  1. minor comments. Please correct in italics all the latin names of bacteria e.g L202 Salmonella, E.coli, etc.
    1. All Latin names of bacteria in the article were reviewed and are correctly written and Italicized. Could not identify the unitalicized bacteria name the reviewer is referring to in line 202 in the original version – all names were italicized (please see lines 212-213).
  1. keywords should reflect the research that was conducted. Why only E.coli O157:H7?
    1. Changed the key word from E. coli O157:H7 to Shiga toxin-producing E. coli and added Salmonella (please see lines 31-32).

Round 2

Reviewer 1 Report

Comments and Suggestions for Authors

Thank you for providing the revised version of the manuscript. 

Comments on the Quality of English Language

It is OK.